# Fibromyalgia Pathophysiology

**DOI:** 10.3390/biomedicines10123070

**Published:** 2022-11-29

**Authors:** Michael Gyorfi, Adam Rupp, Alaa Abd-Elsayed

**Affiliations:** 1Department of Anesthesia, University of Wisconsin School of Medicine and Public Health, Madison, WI 53726, USA; 2Department of Rehabilitation Medicine, University of Kansas Medical Center, Kansas City, KS 66160, USA

**Keywords:** fibromyalgia, pathophysiology, affective spectrum disorders, small fiber neuropathy

## Abstract

This article examines the biological, genetic, and environmental aspects of fibromyalgia that may have an impact on its pathogenesis. Symptoms of fibromyalgia may be related to aberrations in the endogenous inhibition of pain as well as changes in the central processing of sensory input. Genetic research has revealed familial aggregation of fibromyalgia and other related disorders like major depressive disorder. Dysfunctional pain processing may also be influenced by exposure to physical or psychological stressors, abnormal biologic reactions in the autonomic nervous system, and neuroendocrine responses. With more research the pathophysiology of fibromyalgia will be better understood, leading to more logical and focused treatment options for fibromyalgia patients.

## 1. Introduction

The first description of fibromyalgia syndrome (FMS) is found in the nineteenth century. The term “fibrositis”, which Gowers first used in 1904, was in use until the 1970s and 1980s, when a central nervous system-related etiology was proposed [1]. In 1950 Graham described fibrositis as a “pain syndrome” in the absence of a specific organic disease [2]. Then, in the middle of the 1970s, Smythe and Moldofsky created the new term “fibromyalgia” and named the so-called “tender points”—regions of extreme tenderness [3]. The American College of Rheumatology committee’s widely used diagnostic standards, which were only recently modified, were not developed until 1990 [4]. 

FMS is one of the most prevalent disorders that affects the muscles and is characterized by pain, stiffness, and soreness in the muscles, tendons, and joints. Fibromyalgia is in a family of disorders termed the affective spectrum disorders (ASD). ASDs frequently co-occur in both individuals and families and share physiologic abnormalities along with genetic risk factors that may be central to their etiology [5]. The most prevalent ASDs are attention-deficit/hyperactivity disorder, major depressive disorder [MDD], generalized anxiety disorder, obsessive compulsive disorder, bulimia nervosa, panic disorder, posttraumatic stress disorder, premenstrual dysphoric disorder, social phobia, and medical disorders such as irritable bowel syndrome [IBS], migraine, and cataplexy [6].

FMS commonly manifests in young or middle-aged females as chronic widespread pain, stiffness, fatigue, disrupted unrefreshing sleep, and cognitive difficulties. FMS often coexists with a number of other unexplained symptoms, anxiety and/or depression, and functional impairment of daily living activities. Fibromyalgia typically causes broad pain that affects both sides of the body with numerous “tender points”. Despite having incapacitating physical pain, FMS is not accompanied by tissue inflammation, tissue damage, or deformity [7].

The volume and quality of research behind the pathophysiology of fibromyalgia continues to grow. However, there are still healthcare providers that believe that FMS is a mere construct of malingering or attention seeking. Some providers also think that FMS is a purely psychological condition compared to a physical disease. This manuscript will detail abundant research to contradict these outdated thought processes. 

Clinicians should be conversant with the symptoms and signs of fibromyalgia so they can make a timely diagnosis. In the US, the prevalence rate ranges from 6% to 15%, with women experiencing a five times higher frequency than men. The rate of new diagnoses in rheumatology clinics ranges from 10% to 20% compared to 2.1% to 5.7% in non-specialized settings [8]. The symptoms of FMS are often related to stress and fluctuate with time and severity. This review will cover the abnormal pain signaling pathways, genetic predisposition, and environmental triggers for fibromyalgia. 

## 2. Pathophysiology

### 2.1. Abnormal Pain Signaling in Fibromyalgia 

The exact pathophysiological mechanism behind fibromyalgia remains undefined, though it is likely multifactorial in origin including abnormal cortical processing, reductions in inhibitory pain modulatory mechanisms and molecular changes to the pain pathway. What is clear however is that patients with fibromyalgia have increased sensitivity to a variety of stimuli, including mechanical and ischemic pressure, heat, and cold. Normally the source of sensory input (such as a nerve lesion) is discernible in both animal and human models of neuropathic pain, and pain sensitivity is decreased when the source is removed. It is unknown where fibromyalgia patients receive their unique sensory information [9,10]. Because of this, when discussing the pathophysiology of fibromyalgia, the majority of researchers working on the condition use the term central augmentation of sensory input rather than central sensitization. 

Studies have shown that compared to healthy controls, the stimulus intensity required to elicit a pain response in fibromyalgia patients is almost 50% lower [7,11]. This is in addition to various other symptoms including fatigue, insomnia, cognitive impairments, and depression [12]. The pathophysiology of this heightened sensitivity (hyperalgesia and allodynia) has been studied and is thought to involve both impaired central cortical processing as well as dysregulation of the central nervous system at the spinal cord level [13].

A systematic review in 2014 conducted by Cagnie et al. found 22 studies utilizing neuroimaging techniques to evaluate brain activity in patients with FMS. Within their review 8 articles utilized Voxel-based morphometry (VBM) to evaluate brain volume. They collectively found in those with FMS and moderate central sensitization VBM demonstrated physical reductions in gray matter mostly seen in the anterior cingulate cortex and prefrontal cortex, though global gray matter remained unchanged [13]. There did appear to be a temporal correlation between increases in grey matter volume loss and those with long standing FMS [14]. Thirteen studies looked at fMRI images associated with various pain thresholds, and they found increases in cortical blood flow to areas of the brain associated with pain processing with lower stimulation in those with FMS. In addition, they found evidence for a reduction in connectivity in the descending pain-modulation system in FMS patients notably from the anterior cingulate cortex (ACC) to the amygdala, hippocampus, and brain stem [15].

The ACC, periaqueductal grey and rostral ventromedial medulla appear to be large components of the descending pain processing pathway [16]. The descending pain pathway involves the release of endogenous opioids at the spinal level and serotonergic and noradrenergic modulation centrally as it relates to pain perception, both in response to ascending pain signals. In chronic pain states this connection becomes disrupted resulting in decreased opioid release and alteration in serotonin and noradrenergic secretion [17]. 

When incoming A-delta and C-fibers synapse in the dorsal horn they normally release glutamate and substance P which initiates the ascending pain signal through the spinothalamic tract. Opioids normally bind to receptors most commonly mu-opioid receptor (MOR) located in neurons that synapse at this junction. At the spinal cord Opioids act on interneurons that synapse in the dorsal horn both pre and post-synaptically. When activated opioid receptors hyperpolarize ascending fibers and inhibit the release of glutamate and substance P presynaptically. This ultimately reduces the amount of pain signals reaching the brain. In addition, when the pathway is initiated Gaba is reduced in the periaquedutal gray which results in an increase in serotonergic and noradrenergic transmission in the ventromedial medulla thus reducing pain perception [18]. It is hypothesized that repetitive stimulation of C-fibers (such as that seen in chronic pain states like FM) results in apoptosis of the inhibitory opioid and gabanergic interneurons [19]. This reduction in interneurons is associated with increased release of glutamate, substance P, and nerve growth factor at the d second-order neurons and central reductions in opioid and serotonin levels resulting in a “wind-up” mechanism. Wind up is described as a gradual increase in ascending pain signaling along with a reduction in inhibitory descending pain modulation [7,11]. Overtime this mechanism contributes to centralized pain, allodynia, and hyperalgesia as now stimuli that normally would not reach pain thresholds do.

Other proposed mechanisms behind the development of hyperalgesia and allodynia involve microglial cells and the TLR4 receptor. Microglial cells are CNS macrophages or the main defense system of the brain and spinal cord. Normally in a quiescent state, they can become activated by triggers including cell death, peripheral inflammation, and infection. Once activated they undergo morphological changes and release numerous cytokines and other proinflammatory mediators [20]. TLR4 is a receptor found on the outer membranes of microglial cells and they play an integral role in the activation of the cell. TLR4 typically binds to pathogen-associated molecular patterns (PAMPS) and LPS both of which are associated with infection, damaged tissues, or cell death. Upregulation of TLR-4 with repeat chronic stimulation results in proliferation and overactivation of microglial cells. This unregulated activation produces changes not only at the macrolevel affecting pain pathways in the spine and brain but also at the molecular level upregulating gene expression and receptor concentrations [21,22]. 

Other secondary mechanisms that could result in the symptoms experienced by those with FMS include reductions in neurotransmitters, alterations in hormone levels, genetics, and environmental triggers. Serotonin is especially important for regulating mood, depression, and anxiety. Those with FMS have been shown to have reductions in these neurotransmitters when compared to healthy controls. The alterations seen in the pain modulation pathway could account for these reductions in blood serum serotonin levels by reducing Gaba’s effects at the periaqueductal grey [14]. 

Physiological stress can lead to symptoms such as fatigue, decreased work capacity and insomnia. Numerous studies have looked at anomalies of the hypothalamic-pituitary-adrenal (HPA) axis in relation to FMS. Collectively they have demonstrated a hyperactivity of the stress response, albeit the precise nature of these changes has not been clarified. For example, there was a significant relationship between cortisol levels and pain at awakening and an hour after awakening in FMS patients as compared to controls [23,24]. A second study found that patients with fibromyalgia had significantly higher overall plasma cortisol (P 0.001) as well as higher peak and trough levels of cortisol than patients with RA [24]. A study in 2003 evaluated the function of beta-adrenergic receptors on mononuclear cells in 8 FMS females paired with 9 healthy females. They found diminished basal cAMP levels in patients with FMS [25]. In another approach, 16 patients with FMS performed static knee extensions until they became exhausted, plasma catecholamines and ACTH were decreased in comparison to 16 healthy controls in both groups of patients [26]. It is not entirely clear why these aberrant hormone levels exist though studies are ongoing.

Patients with fibromyalgia frequently demonstrate abnormalities in the autonomic nervous system’s (ANS). In addition to reduced microcirculatory responses to aural stimulation, patients with fibromyalgia also displayed attenuated vasoconstriction responses to cold pressor activities [27]. Through the alteration of physiological reactions necessary for efficient stress management, ANS abnormalities may contribute to increased pain and other clinical issues related to fibromyalgia (e.g., increases in blood pressure) and pain inhibition via diminished production of growth hormone (GH) and insulin-like growth factor (IGF-1) [26,27]. 

Innapropriate immune responses and specficially IgG immuoglobulin has demonstrated the ability to cause pain without obvious inflammation or nerve damage [28]. This immune response has been a topic of research for potential causes of conditions characterized by altered nociceptioin that is not fully explained by neuropathic pain mechanisms. One study design, specifically looking at this abnormal immune response, utilized IgG antibodies from patients with potential nociplastic pain syndromes and transfered them to a mouse model. The resulting mice also displayed reduced locomotor activity, decreased strenght, and loss of intraepidermal innervation. This was in contrast to an IgG depleted serum which had no effects on the mice [29]. A separate study examed the link between FMS and satellite glia cells (SGC) binding IgG via immunofluorescence murine cell culture assay. The results demosntrated a correlation between binding of IgG to SGCs and fibromylagia disease severity. These finds show potential for a anti-SGC antibody screening test for FMS to assist personalized treatment options that target autoantibodies [30].

### 2.2. Imaging in Fibromyalgia

Several functional neuroimaging techniques and studies demonstrate quantifiable changes in patients with FMS. One of the first functional neuroimaging techniques to be used to study FMS was the Single-photon-emission computed tomography (SPEC). This method provides a measurement of regional cerebral blood flow (rCBF), which reflects neural activity, throughout the brain following the injection of a radioactive tracer. There seems to be interest in the thalamus. Adigüzel et al. also suggested that the thalamus and basal ganglia are involved. After receiving amitriptyline treatment, he showed increases in rCBF in these areas [31]. Kwiatek et al. observed reduced rCBF in the right thalamus, inferior dorsal pons, and vicinity of the right lentiform nucleus utilizing single photon emission computed tomography [32]. Shokouhi et al. utilized arterial spin labeling to image CBF in 23 FMS patients. They found significant reductions in basal ganglia perfusion [33]. 

Similar to SPECT, positron emission tomography (PET) uses radioactive tracers, but it has higher temporal and spatial resolution. An example of PET utilization in FMS was performed by Wik et al. when he compared rCBF in 8 patients and controls at rest. Patients had greater rCBF in the retrosplenial cortex on both sides compared to controls, but less rCBF in the left frontal, temporal, parietal, and occipital cortices. This may indicate heightened sensitivity to subnoxious somatosensory signaling and an impairment of the typical cognitive processing of pain in fibromyalgia patients [34]. Positron emission tomography with fluorodeoxyglucose (FDG) can be used to detect changes in glucose metabolism. Yunus et al. compared 12 FMS patients and 7 controls and did not detect a change [35]. Contrast this with Walitt et al. who demonstrated a link between the improvement in a thorough treatment program and the rise in metabolic activity in limbic structures [36]. 

Harris et al. used radiolabeled opioid carfentanil in attempts to explain the endogenous opioid system’s apparent paradoxical hyperactivity. He demonstrated that fibromyalgia patients had significantly lower overall mu-opioid receptor binding potential. The areas most significantly involved were the left amygdale, right and left nucleus accumbens, and a tendency toward reduction was also observed in the right dorsal anterior cingulate cortex. These results might be the result of receptor downregulation and occupancy by endogenous opioids released in response to persistent pain [37]. 

Functional magnetic resonance imaging (fMRI), which has higher temporal and spatial resolution than SPECT or PET, is more accurate in both of these areas. In the first fibromyalgia fMRI study fMRI was performed while painful stimulation was applied to 16 patients and 16 controls. First, the stimulation pressure was the same for both groups, and then the intensity of stimulation for the controls was increased to produce a subjective level of pain comparable to that felt by the patients. Only when painful stimulation was present as a second condition did neural activation patterns resemble one another [38]. These results are consistent with a centrally enhanced pain processing model. Cook et al. used fMRI to study how two groups of female subjects—nine patients and nine controls—reacted to painful and nonpainful stimuli. Patients significantly outperformed controls in insular, prefrontal, supplemental motor, and anterior cingulate cortices activity in response to nonpainful stimuli. The contralateral insular cortex was significantly more active in patients than in controls in response to painful stimuli [39]. Other recent studies have hypothesized a role for cortical regions like the right inferior frontal gyrus, insular cortex, supplementary motor area, midcingulate cortex, and right premotor cortex [40]. 

The descending inhibitory pain system’s function appears to be better understood through the use of functional magnetic resonance imaging. In a study by Jensen et al. fMRI was performed while individually calibrated painful pressure was applied to 16 patients and 16 controls. Affective or attentional brain regions, as well as those with sensory connections to the stimulated body area, did not exhibit any differences in activity. The rostral anterior cingulate cortex, a crucial part of the descending pain regulating system, interestingly displayed a diminished response to pain in these individuals [41]. The functional connectivity of the descending inhibitory pain pathways in fibromyalgia patients and controls was compared by Jensen in a later study [15]. The thalamus was more closely connected to the orbitofrontal cortex in controls, as were the bilateral hippocampi, amygdala, brainstem, and rostral ventromedial medulla, regions involved in the pain inhibitory network.

### 2.3. Small Fiber Neuropathies in Fibromyalgia 

Small fiber neuropathy (SFN) is a disease affecting the A-delta or C-fibers of the peripheral nervous system with a proximal, distal, or diffuse distriubtion. SFN typically manifests with pain, sensory disturbances, or autonomic dysfunction. Most SFNs are diagnosed with typical clinical features and a reduced intra-epidermal enrve fiber density via skin biopsy. SFN is brought on by a defect in glucose metabolism, immune dysregulation, gluten sensitivity and celiac disease, monoclonal gammopathy, vitamin deficiency, exposure to toxins, cancer, and unexplained causes [42,43]. 

Fibromylagia syndrom lacks clear defined abnormalities which hinders definitive testing, disease-modifying therapies, and the discovery of causes. One study performed by Oaklander et al. compared 27 FMS patients and 30 controls and found 41% of skin biopsies from subjects with fibromyalgia vs. 3% of biopsies from control subjects were diagnostic for SFPN sugesting that some patients with chronic pain labeled as fibromyalgia have unrecognized or concurrent SFPN [44]. A similar study investigating epidermal nerve fiber density compared calf and thigh 3 mm punch skin biopsies in 41 FMS patients and 47 control subjects respectivly. Their results demonstrated a significantly diminished epidermal nerve fiber density in both anatomical regions in the FMS group which also suggests a roll of small fiber neuropathy in the pathogensis of FMS [45].

Uceyler et al. examined the function and morphology of small nerve fibers in 25 fibromyalgia syndrome patients in this case–control study. Comprehensive neurological and neurophysiological evaluations were performed on the patients. Skin punch biopsies of the lower leg and upper thigh were used to measure intraepidermal nerve fiber density and regenerating intraepidermal nerve fibers, as well as to examine small fiber function using quantitative sensory testing and pain-related evoked potentials. These results were compared with a monopolar depression group and a separte healthy control group matched for age and gender. This study was the first to quantitatively show a reduction in intraepidermal innervation and regneration sparing myelinated nerve fibers along with an elevated cold and warm detection threshold indiciating functional impairment of A-delta and C-fibers [46]. A meta-analysis of the prevalence of small fiber pathology in fibromyaglia was perrofmed in 2019 by Grayston et al. They screened 935 studies and found 8 studies that met their inclusion criteria which provided 222 participants. The results demonstatred a pooled prevalence of SFP in fibromyalgia to be 49% (95% CI: 38–60%) [47]. Evaluation of SPN in FMS enables improved FMS classification, patient care direction, and symptom validation, resulting in better use of resources and outcomes.

An extensive review by Meydan et al. covered the connection between non-coding RNA regulators and diabetic polyneuropathy and their possible connection to FMS. They elaborate on several candidate non-coding RNAs which regulate inflammation, pain-provoking, and metabolic syndrome pathways. Numerous non-coding RNAs were detailed. Although the connection may seem obscure, the authors voiced the importance of understanding the biological pathways involved in polyneuropathies [48]. 

### 2.4. Genetics in Fibromyalgia 

The role of genetics and FMS is not well defined however it is suggested that exposure of a genetically predisposed individual to a host of environmental stressors is presumed to lead to the development of FMS. FMS affects females more often then males in a 2:1 ratio [6]. FMS is characterized by a strong familial aggregation. First-degree relatives of FMS patients are 8.5 times more likely to have the condition than those of RA patients [49]. Evidence for genetic and familial linkgaes was also supported in two large twin survey studies where estimates place heritability at 48–54% [50]. In 2013 a cohort study of 116 families from the Fibromyalgia Family Study underwent the first genome-wide linkage scan for FMS. They found an estimated sibling recurrence risk ratio of 13.6 using a FMS population prevalence of 2 percent further supporting the heritability of FMS [51].

Hundreds of pain-regulated genes have been found, including those for catechol-O-methyltransferase, mu-opioid receptors, GTP cyclohydrolase 1, voltage-gated sodium channels, and GABA-nergic pathway proteins [52]. Several recent candidate based genomic studies have set out to evaluate these pain related genes and found that serotoninergic, dopaminergic, and catecholaminergic gene polymorphisms may play a role in the etiology and pathogenesis of FMS, though these studies have been inconsistent and these polymorphisms are similar to those associated with other comorbid conditions and are not unique to FMS [53]. It has been demonstrated that genetic variations and inheritance mechanisms in genes related to pain play a 50% role in the onset of chronic pain, eluding to the possible relationship between genetic variations and pain response [54]. 

A novel study evaluating genes in Smith et al. conducted the first large scale comparative genomic study on hundreds of individuals with FMS. They evaluated 496 patients with FMS paired with 348 pain free controls was performed in 2012. The Pain Research Panel gene array chip, which tests variations characterizing more than 350 genes known to be implicated in the biologic processes relevant to nociception, inflammation, and mood, was specifically used for genotyping. They found statistically significant diffefrences in three genes including GABRB3, TAAR1 and GBP1 when compared to controls [55]. Alterations to GABRB3 have been shown to procude symptoms of theraml hyperalgesia, tatcile allodynia and att. enuated analgesic responses to Gaba agonists [56]. TAAR1’s role is not entirely clear though some studies suggest they modulate dopaminergic activity [57]. GBP1 is typically induced by interferon and other cytokines and may likely contribute to the symptomotology experienced by chronic inflammatory conditions such as psoriasis and irritable bowel disease. It is not entirely clear it’s role in fibromyalgia [58]. 

In a similar study of a Spanish population, there was a connection between FMS and the “high pain sensitivity” haplotype (ACCG) of the COMT gene. They also found that Inactivating the *MC1R* variants was associated with increased opioid analgesia [59]. Understanding these gene polymorphisms may aid in better subdividing FMS patients and directing a more logical pharmacological strategy. However, despite various genes being linked to FMS there is not a clear significance or innactable course based on their findings. 

### 2.5. Environmental Triggers in Fibromyalgia 

In addition to a genetic predisposition, the environment may play a role in FMS development. Particularly, it has been demonstrated that early life experiences, such as physical trauma and psychosocial stressors. These early insults can affect gene expression and thereby influence the development of FMS [60,61]. Experiences in infancy and childhood have been linked to long-lasting maladaptation in the nociceptive circuitry and an increase in pain sensitivity in adults [62]. 

It has been shown that there is a bidirectional temporal association between depression and FMS, increasing the likelihood that the two will co-occur. The medial orbitofrontal cortex and cerebellum have shown altered gray and white matter morphometry in FMS patients, and the gray matter volume has been linked to the severity of depression and hyperalgesia [63]. It is hypothesized that environmental factors, particularly long-term stress, and traumatic experiences, alter gene expression in a way that affects neurophysiological responses, which in turn affects how both peripheral and central pain are perceived. 

Loggia and colleagues looked at the hypothesis of phycological stressors and their effect on pain scores for patients with FMS compared to healthy controls. They utilized unpleasant odors followed by application of a heat stimulus and measured the perceived unpleasantness. Their findings suggested that psychological and psychosocial stressors disproportionately worsened the perception of pain and discomfort in patients with FMS compared to the controls [64]. 

A retrospective case–control study by Al-Allaf et al. gave a questionnaire to 288 individuals (136 FMS patients and 152 age and sex matched controls from the hospital with various disease states) to assess the prevalence of physical trauma prior to the onset of their respective symptoms. They found 53 (39%) of the FMS patients conveyed significant physical trauma within the six months preceding their FMS symptoms compared to 36 (24%) in the hospital control group [60]. 

Widespread pain has been associated with numerous vitamin deficiencies, however, there is no clear efficacy in replacement therapies. Chronic pain and fibromyalgia have both been linked to thiamine (vitamin B1) deficiency. One case series took female patients with fibromyalgia and gave them high-dose oral thiamine (600–1800 mg/day) from which they experienced significant symptomatic improvements after 20 days of therapy [65]. Although promising, these findings have not been recreated on a reliable basis. 

Fibromyalgia patients frequently experience disordered sleeping, such as nonrestorative sleep, insomnia, early morning awakenings, and poor sleep quality. Patients with fibromyalgia also reported that their pain symptoms got worse after poor sleep quality [66]. One meta-analysis of randomized trials found meditative exercise programs may improve sleep quality in fibromyalgia patients. However, sleep disturbances may have several confounding factors and is difficult to delineate if it is a symptom or a precursor to FMS [67]. 

## 3. Conclusions

The pathophysiology of fibromyalgia is influenced by abnormal pain signaling, genetic predispositions, abnormal neuroendocrine and autonomic system activity, environmental triggers, and sleep disturbances. The pathogenesis of fibromyalgia is not well understood, and the diagnosis remains clinical in practice. The enhanced pain sensitivity and persistence of widespread pain in people with fibromyalgia may be caused by changes in the central processing of sensory input and deficiencies in endogenous pain inhibition. 

Evidence of small fiber neuropathies in patients diagnosed with FMS indicate a connection between the two. Evaluation of SFN in FMS may enable better FMS classification and patient care direction resulting in better use of resources and outcomes. The recent recognition of SFN in a significant subgroup of patients with FMS reinforces the dysautonomia-neuropathic hypothesis and validates fibromyalgia pain. Oxidative stress, and maladaptive immune responses may have a significant impact on the severity of FMS. Although numerous genes abnormalities have been identified in FMS their direct correlation or significance is yet to be seen. These differences could suggest several subgroups if not different disease processes under the FMS diagnosis. 

FMS is apart of the affective spectrum disorders (ASD) which aligns with FMS’s pathogenesis. The development and validation of effective pharmacologic and behavioral treatments for these disorders will be aided by advances in our understanding of the pathophysiology of fibromyalgia and other types of ASDs.

## Data Availability

Not applicable.

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
