# Peer review of "Fibromyalgia Pathophysiology"

_biomedicines, 2022, doi:10.3390/biomedicines10123070_

Round 1

Reviewer 1 Report

I'm deeply disappointed of the value of revised manuscript. Subject is important and there is plenty of data/facts which should be pointed or even mentioned...

The Authors state : "The volume and quality of research behind the pathophysiology of fibromyalgia continues to grow. However, there are still healthcare providers that believe that FMS is a mere construct of malingering or attention seeking. Some providers also think that FMS is a purely psychological condition compared to a physical disease. This manuscript will detail abundant research to contradict these outdated thought processes."

50 out of 66 papers are 10 years or older and 19 from those are older than 20!

Outdated and non-convincing data. Nothing new, not worth to be presented. Refreshed references are necessity, but in such proportions it means new manuscript. 

Final decision is rejection. 

Author Response

Heavily updated with more recent sources as requested. 

Reviewer 2 Report

This comprehensive review covers fibromyalgia, FMG: a disease which is still under dispute, such that its publication is particularly important. The authors collected numerous reports on this disease and wrote a detailed and most informative review, covering both positive and negative views as they should do. These are most valuable notes.

Unfortunately, however, the authors present ample details regarding efforts to find some clues by tests that can always work and are therefore less meaningful than desired. For example, seeking putative links to some genes will always come up with something, which adds a level of doubt to the significance of such findings; likewise, higher incidence of FMG within families may merely reflect higher level of knowhow about this disease at those populations; also, microarray findings are less reliable than needed as the hybridization conditions that are used are the same for many different transcripts, which implies lots of technical errors.

In contrast with these detailed reports of less trustworthy than desired details, I was surprised to miss strong coverage of objective reports. For example, the authors completely ignore complete sets of studies that present significant observations, such as the thin peripheral nerves in FMG patients which could not be identified by chance and were most carefully studied by Prof Nurcan Ucelyer from Wurzburg, Germany. The physiological implications of such data may be immense, and MUST be fully reported. Likewise, animal studies like those of Prof. Svensson from Stockholm are missing, and the recent Meydan clinical-research report is not even mentioned, although their impact among the experts was rather impressive. In conclusion, the authors over-report clinical data that may be less trustworthy than desired and lack research data that were carefully verified and may make a real difference in the field.

Author Response

  1. Discussed the shortcomings of gene linkages.
  2. Covered small fiber neuropathy in detail - Ucelyer. 
  3. Included studies by Svensson and Meydan. 

Reviewer 3 Report

The review by Michael Gyorfi  et al, provides an overview of the pathogenesis of fibromyalgia. The review is clearly written, its original and of interest in its field. 

I recommend that the review be accepted with minor revision:

a)     The authors should better emphasize the conclusions.

b)    The authors should provide an illustrated figure to summarize and simplify about  Pathophysiology .

c)   The literature is poorly updated. Please add recent references. Please referee doi: 10.3390/ijms22083891;10.3390/ijms22126471.

Author Response

  1. Elaborated our thoughts in the conclusion.
  2. We do not have an original figure at this time. 
  3. Updated references and included cited review by Siracusa et al. 

Round 2

Reviewer 1 Report

For sure, to be honest I see improvement. The paper is far away from beeing outstanding and it doesn't go very deep into the merit, however it has been corrected. 

In the perspective of the assessments of other reviewers and the corrections made, my expectations have significantly decreased compared to the previous review and I am now accepting the work for publication.

Reviewer 2 Report

Accept in present form.